# The Healthy Eating Plate Advice for Migraine Prevention: An Interventional Study

**DOI:** 10.3390/nu12061579

**Published:** 2020-05-28

**Authors:** Claudia Altamura, Gianluca Cecchi, Maria Bravo, Nicoletta Brunelli, Alice Laudisio, Paola Di Caprio, Giorgia Botti, Matteo Paolucci, Yeganeh Manon Khazrai, Fabrizio Vernieri

**Affiliations:** 1Headache and Neurosonology Unit, Neurology, Campus Bio-Medico University Hospital, 00128 Rome, Italy; g.cecchi@unicampus.it (G.C.); n.brunelli@unicampus.it (N.B.); m.paolucci@unicampus.it (M.P.); f.vernieri@unicampus.it (F.V.); 2Master’s Degree Course in Food Science and Human Nutrition (SANUM), Campus Bio-Medico University of Rome, 00128 Rome, Italy; mariabravo121193@gmail.com (M.B.); paoladicaprio31@gmail.com (P.D.C.); giorgiabotti@virgilio.it (G.B.); m.khazrai@unicampus.it (Y.M.K.); 3Unit of Geriatrics, Department of Medicine, Campus Bio-Medico University Hospital, 00128 Rome, Italy; a.laudisio@unicampus.it

**Keywords:** migraine, healthy diet, carb

## Abstract

We aimed at evaluating the effect of the Healthy Eating Plate (HEP) education on migraine frequency and disability. At three evaluation times (T-12 = screening, 12 weeks before the intervention; T0 = time of the educational HEP intervention; and T12 = 12-week follow-up), the enrolled subjects underwent assessment of anthropometric and dietary patterns, monthly migraine days (MMDs), and disability scales (Migraine Disability Assessment score (MIDAS), MIDAS A, MIDAS B). The HEP score estimated adherence to dietary advice. We enrolled 204 out of 240 screened migraineurs, of these, 97 patients completed the follow-up. We defined ADHERENTS as patients presenting an increase in HEP scores from T0 to T12 and RESPONDERS as those with a reduction of at least 30% in MMDs. ADHERENTS presented a significant decrease in MMDs from T0 to T12. In particular, RESPONDERS reduced red, processed meat and carb intake compared to NON-RESPONDERS. Reduction in carb consumption also related to a decrease in perceived disability (MIDAS) and headache pain intensity (MIDAS B). Logistic regression confirmed that the HEP score increase and total carb decrease were related to a reduction in MMDs. This study showed that adherence to the HEP advice, particularly the reduction in carb, red and processed meat consumption, is useful in migraine management, reducing migraine frequency and disability. Trial registration: ISRCTN14092914.

## 1. Introduction

Migraine is a chronic neurological disorder with a high social impact on the general population because of its large diffusion and related disability [1]. Migraine pain is the epiphenomenon of a multifactorial cascade of mechanisms whose full understanding remains elusive. Diet has been often implied in its pathophysiology, as most patients with migraine identify skipped meals or some particular foods as precipitating factors [2]. These retrospective self-reports have induced many clinicians to suggest food diaries to individuate and avoid specific food triggers [3]. These dietary regimens rely on the hypothesis that the ingestion of certain staples can directly disrupt homeostasis in neurotransmitter and neuropeptide release. This would be the case of chocolate, wine, aged cheese, processed meats, shelf-stable food, and other products containing vasoactive amines (e.g., histamine, tyramine), nitrites, or monosodium glutamate [4,5]. Another approach based on a possible immune food-related response considers the individualized elimination of food with immunogenic activity by the detection of serum G Immunoglobulins [4]. Finally, other dietary regimens aim at reducing the intake of nutritional compounds (lipids, carbs, and sodium) with hypothetical multiple effects on metabolism. Among these, the ketogenic diet, based on strict carb restriction, has shown promising results in migraine prevention [6], attracting great interest among migraineurs [7]. At the same time, non-scientifically tested diets are diffusely proposed on the web. However, non-controlled diets can have harmful implications, such as electrolyte or vitamin deficits and inadequate nutritional intake. In this scenario, cross-sectional studies have recently shown that the adherence to healthy dietary habits is associated with lower frequency and duration of headaches [8,9].

The Harvard T.H. Chan School of Public Health has proposed a renewed version of the food pyramid: the Healthy Eating Plate (HEP) [10]. According to nutritional guidelines, the Healthy Eating Plate advice provides, in a simple format, detailed guidance to adopt correct eating patterns.

This study aimed at evaluating the effect of education on the Healthy Eating Plate on monthly migraine days (MMDs) and migraine-related disability. We hypothesized that the HEP advice can be of help in the management of migraine and also as an add-on strategy in combination with pharmaceutical therapy.

## 2. Materials and Methods

### 2.1. Study Design

To evaluate the effect of HEP education in patients with migraine, we designed an unblinded longitudinal interventional study consisting of three evaluation times: T-12 = screening, 12 weeks before the intervention; T0 = time of the educational intervention; and T12 = follow-up, after 12 weeks. Our primary endpoint was to establish whether the HEP education reduces MMDs and also evaluate its effect as an add-on to pharmaceutical therapy. Our secondary endpoint was to evaluate whether the HEP advice can alleviate migraine-related disability. The trial was registered as ISRCTN14092914.

### 2.2. Participant Selection

Consecutive patients with migraine treated at our Headache Center were screened for enrolment from March 2018 to September 2019. Migraine was diagnosed according to the International Classification of Headache Disorders [11].

We screened 240 migraine patients. The screening visit included an accurate medical history interview, clinical examination, and anthropometric assessment (height, weight, body mass index—BMI).

Subjects were enrolled if the following criteria were fulfilled:Inclusion criteria: diagnosis of migraine with aura or migraine without aura, age >18-years-old.Exclusion criteria: BMI > 30, cancer, inflammatory bowel disease, celiac disease, type 1 diabetes, chronic renal insufficiency, and other neurological disorders.

Failure to show at control visits and any change in preventive therapy at T0 (dropouts) were additional exclusion criteria for those subjects respecting screening criteria.

Thirty-six patients did not meet the above criteria, of these, 12 subjects, displaying BMI > 30 were referred to nutritional counselling. Eligible patients (204) signed informed consent. 

At T-12, enrolled patients filled a food frequency questionnaire (FFQ) to assess their dietary habits and migraine disability clinical scales concerning the previous three months. They also reported MMDs and painkiller intake in the previous month. All patients received preventive treatment indications as appropriate [12]. At T0, 34 subjects failed to attend the control visit for personal reasons or were unwilling to participate further in the study, the remaining ones underwent again the assessments (FFQ, migraine disability scales, last month MMDs and painkiller intake reporting, BMI). They were all educated about the indications of the HEP by a nutritionist. Of these, 51 patients requiring a change in preventive therapy were considered dropouts. At T12, 22 subjects did not show up. Finally, 97 patients completed all the evaluations and were included in the study (Figure 1). The evaluations at the three times were performed with both the neurologist and the nutritionist. Patients requiring additional visits could receive a neurological consult, but any change in therapy resulted in the exclusion from the study.

The study was approved by our local ethical committee (prot 6.18TS ComET CBM).

### 2.3. Anthropometric Measurements

Subjects’ weight and height were measured at T-12, T0, and T12. Patients were weighed while wearing light clothes (i.e., no sweaters, jackets, or belts) and without shoes to avoid possible confounders for repeated measurement (including seasonal differences). Weight was measured to the nearest 0.1 kg. Height was measured to the nearest 0.1 cm using a stadiometer, while the person was in a standing position with shoes removed, the shoulders were relaxed while looking straight ahead with the Frankfurt plane horizontal (scale and stadiometer, Fazzini, Milan, Italy) [13]. BMI was calculated from the height and weight data, using the “weight (kg)/height^2^ (m)” equation.

### 2.4. Migraine Attack Frequency and Disability Assessment

Monthly migraine days, painkiller intake, and disability scales were assessed at T-12, T0, and T12. All patients were treated at our Headache Center and had been previously educated about correctly recording their daily headache diaries for clinical purposes. We collected data on MMDs and painkiller consumption from headache diaries. Migraine disability was assessed by a validated Italian version of the Migraine Disability Assessment score (MIDAS) questionnaire [14]. MIDAS is a semi-quantitative score measuring migraine days (MIDAS A), pain intensity (MIDAS B), and days of absence or reduced activity at work or in the household due to headaches (MIDAS score) in the previous three months.

### 2.5. Dietary Assessment and Education

Food intake was assessed in all subjects at T-12, T0, and T12. Dietary patterns over the previous 12 weeks were evaluated using a modified version of a semi-quantitative food frequency questionnaire (FFQ) validated in the Italian population [15]. This questionnaire includes a list of 110 food items. For the aim of the current study, i.e., adherence to the HEP advice, we added in the “cereals and bread” section further items, namely: spelt, barley, and whole-wheat breakfast cereals with a total count of 113 food items. Before completing the FFQs, subjects were instructed on “serving size” amounts according to the Italian Society of Human Nutrition [16]. The FFQs were self-administered; however, a nutritionist was available in case participants felt insecure about compilation.

While filling the questionnaire, patients were asked to reply to some additional questions:

At T-12, T0, and T12:Q1-WATER: “How many liters of water do you drink daily?”At T0
Q2-EXPECTANCIES: “Do you think that a healthy diet can help improve headaches?”At T12
Q3-SELF REPORTED ADHERENCE “Were you able to follow the healthy eating plate advice?”Q4-SUGAR: “Were you able to reduce daily sugar consumption?”Q5-SALT: “Were you able to reduce salt use as flavor enhancer?”Q6-EXERCISE: “Were you able to exercise at least 30 min a day”Q7-EXPERIENCE: “In your experience, do you think that the Healthy Eating Plate advice helped you improving headaches?”

Subjects were required to answer “yes” or “no” to questions Q2–Q7. 

At T0, after they had filled the questionnaires, a nutritionist, instructed all subjects in an individual session, about HEP. The HEP education lasted at least 15 min and was considered as complete only when the subject declared to have fully understood the educational content. All subjects received a colored printed image of the HEP as well as the Italian written indications [10].

### 2.6. Data Processing

Food groups are summarized in Appendix A. Food intake frequency was calculated as weekly consumption, while water intake was based on daily drinking. For the statistical analysis, we grouped the sum of refined cereal, potato, and high-carb breakfast and snack intake as TOTAL CARBS, the sum of whole-grain breakfast and cereal intake as TOTAL WHOLE-GRAINS, the sum of fish and legume intake as HEALTHY PROTEINS, and the sum of white meat, egg, and cheese intake as OTHER PROTEIN.

To assess adherence to the HEP diet, we created a score (HEP score) ranging from 0 to 10 where one point was scored if each HEP indication was followed according to the results of the FFQ (Table 1).

We defined ADHERENTS as patients presenting an increase in HEP scores from T0 to T12.

We calculated variation from T0 to T12 as the absolute difference in BMI, food intake frequency, HEP scores, disability measures (MIDAS score, MIDAS A, and MIDAS B), last-month MMDs, and painkiller intake. Since most patients were already on pharmacological preventive treatment, we considered RESPONDERS as subjects who achieved at least a 30% reduction in MMDs from T0 to T12. The reduction in MMDs is to be intended as a further reduction after the effect of the ongoing preventive treatment, unchanged throughout the study. This is in line with other migraine clinical settings where the expected benefit is mild [17].

### 2.7. Statistical Analysis

Statistical analyses were performed using SPSS 25.0; SPSS Inc., Chicago, IL, USA. Differences were considered significant at the *p* < 0.050 level. The sample size was based on our previous experience with this design and further amplified [18]. Data distribution was assessed by the Kolmogorov–Smirnov test. Data of continuous variables are presented as mean values ± standard deviation (SD). Median values with inter-quartile ranges (IQr) were provided for non-normally distributed variables. Analysis of variance (ANOVA) for normally distributed variables was performed according to RESPONDER or ADHERENT status; otherwise, the nonparametric Mann–Whitney U test was adopted. The two-tailed Fisher exact test was used for dichotomous variables. To assess changes over time, paired t-test or Friedman analysis of rank were adopted. Multivariable linear regression analysis (forced entry) was used to assess the association of the last-month changes in MMDs with age, sex, HEP score, Q6-EXERCISE response, and all those variables which differed significantly (*p* < 0.050) in RESPONDERS compared with NON-RESPONDERS.

## 3. Results

Table 2 summarizes demographic and anthropometric measures and migraine disability scales showing that the whole group presented a reduction in BMI and MMDs in the three months before the last evaluation (MIDAS A).

Preventive therapies were prescribed at T-12 and continued at the same dose until T12 in 74.6% of our patients and distributed as follows: 22.2% tricyclic antidepressants (amitriptyline), 14.1% beta-blockers (propranolol), 10.1% calcium-antagonists (flunarizine), 9.1% botulin toxin, 7.1% antiepileptic drugs (5.6% topiramate, 1.5% valproate).

Table 3 condenses food consumption along the three evaluation times: dietary habits differed only for a few food groups from T0 to T12.

A positive response to Q2-EXPECTANCIES was obtained in 62.9% of cases, in 53.6% to Q3-SELF-REPORTED ADHERENCE, 77.3% to Q4-SUGAR, 73.2% to Q5-SALT, 50.5% to Q6-EXERCISE, 60.8 to Q7-EXPERIENCE. Twenty-three (23.7%) patients were classified as ADHERENTS. From T0 to T12, ADHERENTS presented a significant reduction in MMDs compared with NON-ADHERENTS (*p* = 0.007, Figure 2A) and a decrease in monthly painkiller intake albeit non-statistically significant (*p* = 0.063, Figure 2B).

Moreover, ADHERENTS were more frequently RESPONDERS (*p* = 0.012). No difference was observed for T0–T12 variation in MIDAS (*p* = 0.951), MIDAS A (*p* = 0.086), MIDAS B (*p* = 0.166).

Table 4 evidences anthropometric measures and food consumption frequencies in RESPONDERS compared with NON-RESPONDERS, highlighting that RESPONDERS significantly presented a reduction in red and processed meat and TOTAL CARB intake, while no difference was observed for BMI.

Spearman correlation showed that the change in HEP scores was related to MMD variation (ρ = −0.290 with *p* = 0.004). Finally, changed TOTAL CARB consumption was related to MMD (ρ = 0.243, *p* = 0.016) and painkiller intake variation (ρ = 0.288, *p* = 0.004) as well as to changes in perceived disability (MIDAS score, ρ = 0.372, *p* <0.0001) and pain intensity (MIDAS B, ρ = 0.220, *p* = 0.033). To note, TOTAL CARB intake and HEP score variations were not related to each other (*p* = 0.137).

TOTAL CARBS, red and processed meat consumption, and HEP score changes were entered in a logistic regression model corrected for sex and age, ongoing preventive therapy, and response to Q6-EXERCISE to assess their influence on absolute changes in MMDs. The logistic regression confirmed the main effect of the HEP score increase and TOTAL CARB decrease on reduction in MMDs (Table 5).

No interaction was observed between the RESPONDER status and the answers to Q2-EXPECTANCIES, Q5-SALT, Q6-EXERCISE. A positive response to Q3-SELF-REPORTED ADHERENCE (*p* = 0.032), Q4-SUGAR (*p* = 0.037), and Q7-EXPERIENCE (*p* = 0.048) were more often observed in the RESPONDER group. Conversely, a positive response to Q3-SELF-REPORTED ADHERENCE was not significantly related to ADHERENT status (*p* = 0.242).

Preventive therapies were equally prescribed in ADHERENTS compared with NON-ADHERENTS (*p* = 0.805) and in RESPONDERS compared with NON-RESPONDERS (*p* = 0.187).

## 4. Discussion

This study shows for the first time that promoting healthy eating habits might be beneficial in migraine prevention. Moreover, the reduction in carbs, red and processed meat consumption seems to specifically play a significant role in decreasing migraine frequency.

Different dietary regimes have been proposed so far as an adjunctive strategy in migraine management. Most of these diets are based on the avoidance of foods hypothesized to activate the attacks by influencing the plasma levels of molecules (such as Calcitonin Gene-Related Peptide, nitric oxide (NO), and serotonin) largely involved in migraine pathogenesis, or by affecting different aspects of brain homeostasis such as neuronal energy efficiency, excitability, and inflammation (also with an immunogenic response) or platelet aggregation [19]. Similarly, the ketogenic diet, which can exert a pharmacological action on cortical excitability by increasing the mitochondrial energy efficiency, activating GABAergic pathways and suppressing neuroinflammation, is based on the nearly complete elimination of carbs [20,21]. The Healthy Eating Plate, in line with nutritional guidelines, does not prohibit any food; in contrast, it encourages varied nutrient consumption even though recommending a restriction for certain staples. A healthy diet also seems to have beneficial effects in other neurological disorders [22,23]. The mechanisms subtending these positive influences of a healthy diet on disease history can be numerous.

First, the Healthy Eating Plate recommends whole partly processed carbs, with a low glycemic index. The role of glucose metabolism in migraine physiopathology has been largely investigated, although a unique pathogenetic hypothesis is difficult to draw [24]. Cerebral metabolism cannot rely on glycogen storage, being in constant need of oxygen and glucose supply from systemic circulation. A typical clinical example is fasting-triggered headaches. Hypoglycemia produces glucose shortage in the brain and catecholamines release resulting in sympathetic activation. Both mechanisms may favor migraine onset. As an apparent paradox, a low-glycemic-index diet may be of benefit in migraine prevention [25]. Indeed, the reduction in blood glucose oscillation allows the stability of glucose regulating hormones that in turn have been involved in migraine pathogenesis [24]. A study on male rats observed that insulin and glucagon can alter the transmission of nociceptive inputs in the trigeminal–cervical complex suggesting another important potential neurobiological link between migraine and impaired metabolic homeostasis [26]. More recently, Kilic and colleagues reported that inadequate brain glycogen can increase spreading depression susceptibility [27]. In line with these hypotheses, in our cohort, a net improvement in migraine attack frequency and related disability was strongly related to carb restriction. Moreover, RESPONDERS reported having reduced added sugar more frequently than NON-RESPONDERS. However, the reduction in carb intake is not the only aspect justifying the beneficial effect of the HEP diet. To note, the change in the HEP score did not relate to variation in carb consumption. Furthermore, while ADHERENTS presented a significant reduction in MMDs both statistically and clinically (Figure 2) in the month preceding T12, disability scale scores did not improve. Conversely, reduction in carb intake was related to the reduction in disability scales, suggesting that while carb intake can have a rapid impact on migraine, the HEP diet as a whole requires a longer time to achieve its beneficial effects. To explain this, we can hypothesize that carb intake, as above described, has a direct effect on neural homeostasis, while the HEP diet may act through intermediate and more complex pathways.

One possible mechanism is the increase in nutrients with anti-inflammatory properties (fish, nuts, vegetables, and fruits) and the reduction in food with inflammatory potentials (red meat, dairies), together with adequate hydration proposed by the HEP diet. This more favorable balance together with a reduction in highly glycemic food may also have benefited the wealth of gut microbiota. The gut–brain axis is the topic of an increasing number of studies. It has been implied in a variety of neurological and psychiatric disorders. A clear relation between gut microbiota and migraine is still to be established; however, its impact on the central and peripheral nervous systems supports a role in the migraine physiopathology [28].

Although only half of the patients declared to have followed our nutritional advice, a positive reply was more frequently observed in RESPONDERS. Interestingly, while we observed no relationship between an expected connection between migraine and diet at the beginning of the study (Q2-EXPECTANCIES), at the end of the study subjects in the responder’s group affirmed more frequently that the HEP diet was of help for their migraine. The low adherence to the HEP advice and the relatively high number of dropouts were the only discouraging observations in this study. Beyond the self-reported adherence to the diet, the calculated HEP score, considering all enrolled patients, was low among all the evaluation times and paradoxically decreased from T0 to T12. Nevertheless, ADHERENTS presented a significant impact on MMDs (Figure 2). Even if the HEP diet is mainly focused on the quality rather than the quantity of food, patients may have encountered some difficulties in the everyday preparation of certain meals (especially vegetables). On the other side, at T12 subjects presented an overall reduction in BMI, an increase in whole-grain cereal consumption and water intake and a reduction in red and processed meat, refined cereal products, and sweetened beverages, suggesting that education on a healthy diet somehow increased their dietary awareness. To note, in our cohort weight loss was not associated with migraine improvement (Table 5). Similarly, salt reduction, which was self-reported in nearly 75% of cases, and exercise, seemed not to exert any effect in our cohort. Moreover, most patients were on pharmacological preventive therapy at the beginning of the study, which was maintained steady along with visits. Nevertheless, it did not justify the observed migraine alleviation in ADHERENT patients as demonstrated by regression analysis.

The lack of a control group can be pointed out as the main limitation of this study. In this pivotal study, we aimed at offering HEP advice as a complementary approach to standard pharmacological care. Nevertheless, the longitudinal design provided an assessment of the three months before the HEP education, allowing a within-subject comparison in order to reduce possible confounders related to ongoing pharmacological therapy, which was also included as an independent variable in multiple regression analysis. Additionally, we assessed the effect of the HEP advice on migraine impact in patients who were more adherent compared to those who did not modify their diet. Another limitation of the study is the observed high rate of dropout due to the clinical indication to modify the prophylaxis drug before the educational intervention, screening failure, or incomplete follow-up. However, we observed the most consistent sample reduction preceding the educational intervention. For the above limitations, a controlled randomized trial on a wider adherent population is imperative to confirm the results of our study.

## 5. Conclusions

In conclusion, our study suggests that an inclusive healthy diet can be beneficial in migraine management. General practitioners and headache specialists could also consider advising patients on a healthy diet, among other regimens, in migraine patients without comorbidity, according to individual preferences. This may have implications for the best migraine care and long-term benefits for health.

## Figures and Tables

**Figure 1 nutrients-12-01579-f001:**
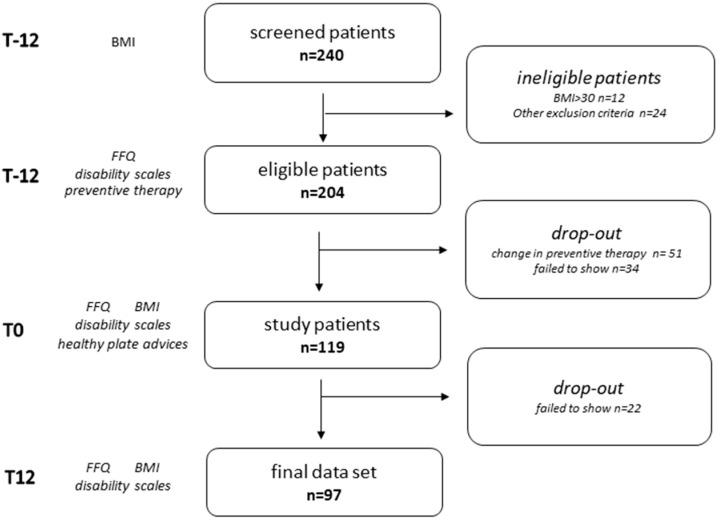
Study design and population at the three evaluation times. T-12: screening, 12 weeks before the intervention; T0: time of the educational intervention; T12: follow-up, after 12 weeks; FFQ: food frequency questionnaire; BMI: body mass index.

**Figure 2 nutrients-12-01579-f002:**
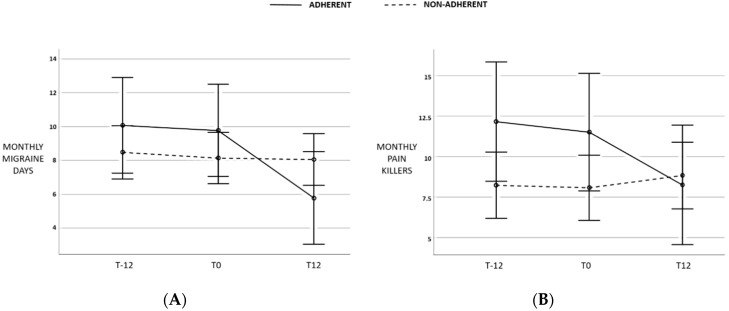
Changes (**A**) in monthly migraine days (MMDs) and (**B**) painkiller intake in ADHERENTS compared with NON-ADHERENTS in the month preceding T12. Bars indicate 95% confidence intervals.

**Table 1 nutrients-12-01579-t001:** The Healthy Eating Plate score.

Item	Yes	No
Water per day ≥2 L	1	0
At least 2 servings of seasonal fresh fruits per day	1	0
At least 3 servings of vegetables per day	1	0
No more than 2 servings of milk/dairy per day	1	0
At least 8 servings per week of whole grains in the mean AND a ratio of whole grains/refined grain and potato consumption >1	1	0
Having for breakfast whole grains cereals or biscuits at least 5 times per week	1	0
Less than 3 servings a day of fat dressing AND a ratio of healthy vegetable oils/partially hydrogenated oils and animal fats >3	1	0
Having no more than 1 sweetened beverage (including packaged fruit juice) per week	1	0
Having 1–2 serving of nuts as a snack/day	1	0
**Subtotal**		
Healthy protein sub-score		
Cheese no more than 1 serving per week	1	0
1 or 2 eggs per week	1	0
Fresh fish (including shellfish) at least 2 servings per week	1	0
Beans at least 2 servings per week	1	0
Poultry: 2 servings per week	1	0
Red meat or processed meat once a week or less	1	0
Protein subtotal		
Protein Score	Protein subtotal/6	
Total	Subtotal + Protein Score	

**Table 2 nutrients-12-01579-t002:** Demographic, anthropometric measures, and migraine disability scales.

n = 97	T-12	T0	T12	*p*
Sex, F (%)	84.5	-	-	
Age, years (mean, SD)	42.08 (12.93)	-	-	
**BMI, kg/m^2^ (median, IQr)**	**23.2 (5.30)**	**23.21 (5.65)**	**23.14 (5.18)**	**0.009**
Last month migraine days (median, IQr)	7 (8)	7 (7)	6 (7.5)	n.s.
Last month pain killer intake (median, IQr)	6 (8.5)	6(8)	6 (9,5)	n.s.
**MIDAS A (median, IQr)**	**18 (20)**	**18 (19)**	**15 (19.5)**	**0.001**
MIDAS B (median, IQr)	7 (2)	7(3)	7 (2)	n.s.
MIDAS score (median, IQr)	14 (26)	13 (22)	15 (34.75)	n.s.

IQr: interquartile range; MIDAS: Migraine Disability Assessment score, MIDAS A: Migraine Disability Assessment A score; MIDAS B: Migraine Disability Assessment B score. In bold are evidenced significant changes of variables along the evaluation times.

**Table 3 nutrients-12-01579-t003:** Food consumption frequency and the Healthy Eating Plate score along evaluation times.

n = 97 (Median, IQr)	T-12	T0	T12	*p*
**Water intake, L/day**	**1.00 (0.50)**	**1.00 (0.5)**	**2.00 (1.00)**	**<0.0001**
Fresh fruits, serving/week	7 (10.5)	7 (10)	7 (8.5)	n.s.
Vegetables, serving/week	11 (8)	10 (8)	10 (7)	n.s.
Milk and diary, serving/week	4 (7.5)	7 (8)	6 (7)	n.s.
**Whole grain cereals, serving/week**	**1 (5.5)**	**1 (6)**	**3.5 (7)**	**0.001**
**Refined cereals and potatoes, serving/week**	**12 (12)**	**12 (11.5)**	**9 (9)**	**<0.0001**
**Whole grain breakfast, serving/week**	**0 (3.5)**	**0 (2)**	**2 (6.5)**	**0.002**
High-carb breakfast and snacks, serving/week	2 (6)	2 (7)	2 (69	n.s.
Vegetable oils, serving/week	14 (6.25)	14 (6)	14 (7)	n.s.
Animal fat and margarine, serving/week	0 (1)	0 (1)	0 (0)	n.s.
Cheese, serving/week	4 (4)	4 (5)	4 (6)	n.s.
Eggs, serving/week	1 (1)	1 (1)	1 (1)	n.s.
Legumes, serving/week	1 (1)	1 (1)	1 (1)	n.s.
Fish, serving/week	1 (1)	1 (1)	1 (1)	n.s.
White meat, serving/week	2 (3)	2 (3)	2 (3)	n.s.
**Red and processed meat, serving/week**	**4 (3)**	**4 (4)**	**3 (2)**	**<0.0001**
**Sweetened beverage, serving/week**	**1 (4)**	**1 (2)**	**0 (2)**	**0.006**
Nuts/serving/week	1 (4)	1 (4)	1 (4)	n.s.
Alcohol/serving/week	0 (2)	0 (2)	0 (2)	n.s.
**Healthy Eating Plate score**	**3.5 (1.33)**	**4.67 (2.05)**	**4.33 (1.84)**	**<0.0001**

In bold are evidenced significant changes of variables along the evaluation times.

**Table 4 nutrients-12-01579-t004:** Changes in food group weekly intake in RESPONDERS compared with NON-RESPONDERS.

Food Group (Median, IQr)	RESPONDERS (n = 33)	NON-RESPONDERS (n = 64)	*p*
Water intake, L/day	0 (0.5)	0.8 (0,5)	n.s.
Fresh fruits, serving/week	0 (5)	0 (3)	n.s.
Vegetables, serving/week	14 (13)	10 (17)	n.s.
Milk and diary, serving/week	0 (1)	0 (2)	n.s.
TOTAL WHOLE-GRAINs, serving/week	6 (8)	1.5 (5.75)	n.s.
**TOTAL CARBs, serving/week**	**−4.5 (10)**	**−3 (12.75)**	**0.015**
Vegetable oils, serving/week	0 (10.25)	0 (0.75)	n.s.
Animal fat and margarine, serving/week	0 (1)	0 (0)	n.s.
**Red and processed meat, serving/week**	**−1.5 (3)**	**−0.5 (3)**	**0.033**
HEALTHY PROTEINs, serving/week	−0.5 (7)	0 (3)	n.s.
OTHER PROTEINs, serving/week	1 (6.75)	2 (6.75)	n.s.
Sweetened beverage, serving/week	−0.5 (1.25)	0 (1)	n.s.
Nuts, serving/week	0 (2)	0 (1)	n.s.
Alcohol, serving/week	0 (1.25)	0 (1.75)	n.s.

In bold are evidenced significant difference between RESPONDER and NON-RESPONDER groups.

**Table 5 nutrients-12-01579-t005:** Logistic regression of MMD changes on age, sex, BMI, TOTAL CARB intake, red and processed meat intake, and healthy plates score T0–T12 variations, ongoing preventive therapy, and physical activity.

	B	Sign.	95% C.I.	
			Lower Limit	Upper Limit
Age	−0.019	0.715	−0.125	0.086
Sex	1.520	0.377	−1.881	4.921
BMI	−0.370	0.076	−0.778	0.039
**Healthy Eating Plate score**	**−1.118**	**0.025**	**−2.093**	**−0.142**
Red and processed meat	−1.118	0.025	−2.093	−0.142
**TOTAL CARBS**	**0.159**	**0.028**	**0.018**	**0.301**
Q6-EXERCISE	0.680	0.592	−1.830	3.190
Preventive therapy	1.956	0.130	−0.589	4.501
Constant	4.401	0.384		

In bold are evidenced variables significantly associated with MMD changes.

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
