# Peer review of "The Healthy Eating Plate Advice for Migraine Prevention: An Interventional Study"

_nutrients, 2020, doi:10.3390/nu12061579_

Round 1

Reviewer 1 Report

Comments to the Authors

Manuscript ID: Nutrients-788511. “The Healthy Eating Plate advice for Migraine prevention: an interventional study”.

The article is interesting, original, well structured and can be improved.

Introduction

The introduction is brief and it is poorly documented especially regarding studies related to diet and migraine.

Material and Met

The study lacks a control group and this aspect is important in interventional studies.

Very limited information is included in the study regarding pharmacological treatment

The authors indicate that 34 patients were failed from the study but they do not justify why.

I have a question: is there any reason the authors applied in trail research Food pyramid: the Healthy Eating Plate (HEP) instead of the mediterranean diet pyramid since the study population is Italian and the research is carried out in Italy

Results

Figures 2 and 3 could be eliminated since they do not provide anything new that is not included in the tables. Respect to table 5, Odd ratios should be included

Conclusion

I consider that the research is very interesting and is a great subject for study, but more research should be done to propose the recommendations indicated by the authors.

Author Response

Manuscript ID: Nutrients-788511. “The Healthy Eating Plate advice for Migraine prevention: an interventional study”.

The article is interesting, original, well structured and can be improved.

Introduction

The introduction is brief and it is poorly documented especially regarding studies related to diet and migraine.

R: we thank the reviewer for this suggestion. In the new version of the manuscript, we described the main concepts of diet regimens proposed for migraine so far. (page 1, lines 32-44)

Material and Met

The study lacks a control group and this aspect is important in interventional studies.

R: We are aware that is the main drawback of the study. In the discussion section, we put more emphasis on this point explaining the reasons that led us to this design. At this stage, we preferred to perform a within-subject rather than an across-subject comparison. Being the first study investigating the possible effect of HEP advice in a real-life setting, an across-subject comparison would have implied a strict cross-matching of baseline migraine frequency and disability, the proportion of different pharmacological therapies, and adherence to HEP advice.  (page 9, lines 302-313)

Very limited information is included in the study regarding pharmacological treatment

R: in the revised manuscript we specified the proportion of specific drug prophylaxis and better explained drug prescription and treatment length. (page 6, lines 181-184)

The authors indicate that 34 patients were failed from the study but they do not justify why.

R: We specified that 34 patients did not show up at the T0 visit (i.e. before intervention), either since they just couldn’t come or they were no longer interested in participating in the study (page 2, lines 84-85)

I have a question: is there any reason the authors applied in trail research Food pyramid: the Healthy Eating Plate (HEP) instead of the mediterranean diet pyramid since the study population is Italian and the research is carried out in Italy

R: We selected the Healthy Eating Plate (HEP) as it is mainly based on recommendations of Mediterranean diet but at the same time it provides an easier tool to graphically illustrate the proportion of nutritional compounds for each meal. Also, it gives more emphasis on whole-grain compared to refined cereals.

Results

Figures 2 and 3 could be eliminated since they do not provide anything new that is not included in the tables. Respect to table 5, Odd ratios should be included

R: we thank the reviewer for this suggestion. We erased figure 3 as it is redundant with the presented results. Concerning figure 2, we would keep it as in the tables are described the data from the whole group, while the figure compared adherent with non-adherent subjects.

In table 5 (and statistical analysis description), we made a mistyping between logistic and linear. In a previous version of the manuscript, we had performed a logistic analysis, comparing responders with non-responders. We later preferred to carry out a linear regression having MMDS as the dependent variable. Thank you for bringing it up, we have corrected this typo. (page 5, lines 170-172)

Conclusion

I consider that the research is very interesting and is a great subject for study, but more research should be done to propose the recommendations indicated by the authors.

R: in the discussion section we put more emphasis on the fact that the results from this study need confirmation. (page 9, lines 312-317)

Reviewer 2 Report

The authors present a longitudinal study on the effect of dietary modifications (the healthy eating plate) for migraineurs. The topic is of major interest for many patients, however, there are some points to discuss with this manuscript/study.

Abstract: “we enrolled 204 out of 204 screened migraineurs…”, please correct into 204 of 240

Methods: 240 patients were screened, 204 included but only 97 are completers. This is one major weakness of the study. Patients were excluded due to changes of their preventive medication. The use of preventive medication is a second major confounder of the study. Some preventives increase weight (amitriptyline, flunarizine) others may decrease weight and hunger (topiramate). The preventives used during the study course are not reported. Frequency of visiting a doctor or the headache center besides the study visits are not reported. The intervention itself (counseling about the healthy eating plate concept) is not described in detail (how long? Single session? Group session? Only written material?). Monthly migraine days are not captured by a headache diary. The study is missing a control group which is strongly recommended (a minimum requisite will be comparing to a standard of care group.

Results: BMI change is significant (.009). Is this really correct (BMI changes from 23.2 to 23.21 and 23.14 in n=97, this do not look significant) However a BMI change of 0.1 is clinically totally irrelevant.

Author Response

The authors present a longitudinal study on the effect of dietary modifications (the healthy eating plate) for migraineurs. The topic is of major interest for many patients, however, there are some points to discuss with this ma

Abstract: “we enrolled 204 out of 204 screened migraineurs…”, please correct into 204 of 240

R: Thank you very much for having underlined the typo.          

Methods: 240 patients were screened, 204 included but only 97 are completers. This is one major weakness of the study. Patients were excluded due to changes of their preventive medication.

R: We agree that the big loss of patients in this longitudinal study is the main concern. Indeed, we have expected a high percentage of lost subjects to follow-up, this is the reason why we have screened many patients. This may be in part due to the study length.  In the revised version of the manuscript, we commented on this point as a limitation of the study in the discussion section. (page 9, lines 310-313)

The use of preventive medication is a second major confounder of the study. Some preventives increase weight (amitriptyline, flunarizine) others may decrease weight and hunger (topiramate). The preventives used during the study course are not reported.

R: We specified for each class of preventives, the absolute percentage of the specific molecules. Preventive drugs were entered in the regression analysis and did not influence MMDs change from T0 to T12. Indeed, we have excluded any patient requiring dose or drug changes along the study to maintain a stable therapy for 6 months. We thus believe that changes observed at T12 compared to T0 can be attributable to the dietary intervention rather than to a substantial change in drug effectiveness alone after 6 months. (page 5, lines 178-181)

Frequency of visiting a doctor or the headache center besides the study visits are not reported.

R: we clarified this point. We designed an unblinded longitudinal interventional study consisting of three evaluation times: T-12=screening, 12 weeks before the intervention; T0=time of the educational intervention; and T12= follow-up, 12 weeks after. Patients requiring additional visits could receive our neurological advice, but any change in therapy resulted in the exclusion from the study. (page 2-3, 90-92).

The intervention itself (counseling about the healthy eating plate concept) is not described in detail (how long? Single session? Group session? Only written material?).

R: We better described the interventional education in the new version of the manuscript. (page 4, 136-139)

Monthly migraine days are not captured by a headache diary.

R: As stated in the “Migraine attack frequency and disability assessment” section, Monthly migraine days, painkiller intake and disability scales were assessed at T-12, T0, and T12. All patients were treated at our Headache centre and had been previously educated about correctly recording their daily headache diaries for clinical purposes. We collected from headache diaries MMDs and painkiller consumption. (page 3, lines 105-112).

The study is missing a control group which is strongly recommended (a minimum requisite will be comparing to a standard of care group.

R: We are aware that it is the main drawback of the study. In the discussion section, we put more emphasis on this point explaining the reasons that led us to this design. At this stage, we preferred to perform a within-subject rather than an across-subject comparison. Being the first study investigating the possible effect of HEP advice in a real-life setting, an across-subject comparison would have implied a strict cross-matching of baseline migraine frequency and disability, the proportion of different pharmacological therapies, and adherence to HEP advice.   (page 9, lines 302-313)

Results: BMI change is significant (.009). Is this really correct (BMI changes from 23.2 to 23.21 and 23.14 in n=97, this do not look significant) However a BMI change of 0.1 is clinically totally irrelevant.

R: we ran again the statistics (Friedman analysis of rank), which confirmed the statistical significance of the BMI difference along times. Since this variable did not have a normal distribution in our cohort, the table expresses it as median with IQr. The mean values of BMI were at T-12=23,85, at T0=23,96 and T12=23,61.

Round 2

Reviewer 1 Report

Some suggestions have been considered and the article has been improved. However the control group is interesting in this type of studies

Reviewer 2 Report

The authors adress the points of the reviewer in an adaequate manner. The lack of a control group is still the major topic but could not changed after finishing the project.